# Characteristics of Tidal Discharge and Phase Difference at a Tidal Channel Junction Investigated Using the Fluvial Acoustic Tomography System

**Mochammad Meddy Danial [1,2], Kiyosi Kawanisi [1,*] and Mohamad Basel Al Sawaf [1]**

[1]   Department of Civil and Environmental Engineering, Graduate School of Engineering, Hiroshima University, 1-4-1 Kagamiyama, Higashi Hiroshima 739-8527, Japan; meddydanialstmt@gmail.com (M.M.D.); mbasel@hiroshima-u.ac.jp (M.B.A.S.)

[2]   Faculty of Engineering, Tanjungpura University, Pontianak 78124, Indonesia

*   Correspondence: kiyosi@hiroshima-u.ac.jp; Tel.: +81-82-424-7817

**Abstract:** This study investigates the tidal discharge division and phase difference at branches connected to a channel junction. The tidal discharge at three branches (eastern, western, and northern branches) was continuously collected using the fluvial acoustic tomography system (FATS). The discharge asymmetry index was used to quantify the flow division between two seaward branches (eastern and western branches). The cross-wavelet method was applied to calculate the phase difference between the tidal discharge and water level. The discharge asymmetry index shows that the inequality of flow division is obviously prominent during the spring tide duration, where the eastern branch has the capability to deliver greater amounts of subtidal discharge, approximately 55–63%, compared with the western branch. However, the equality of flow division between the eastern and western channels can be observed clearly during the neap tide period. The wavelet analysis shows that the phase difference at the western branch is higher than at the eastern branch, because the geometry of the western branch is more convergent than that of the eastern branch. Accordingly, the amplitude of the tidal wave at the western branch is more magnified compared with that at the eastern branch. Moreover, the phase difference at the northern branch is greater than at the two seaward branches, implying that the phase difference is slightly increased after passing through the junction into the northern branch.

**Keywords:** tidal discharge; phase difference; tidal channel junction; flow division; fluvial acoustic tomography; wavelet analysis

## 1. Introduction

One of the most important features in a tidal channel network is the bifurcations, which are mostly located in the middle section of a delta. In the tidal channel network, there are many branches/ junctions that are commonly bifurcated asymmetrically [1]. Because of the different geometrical shape of branches, the behavioral pattern and magnitude of discharge in the first branch are different from the second branch. In a tidal channel junction where there are three channels connected, the tides that propagate upstream in the channels affect each other, and the tidal energy can propagate in two directions. As a result, the magnitude of discharge and its phases in three tidal channels are not similar [2,3].

An investigation of hydrodynamics at the tidal channel junction related to the interaction between the tidal wave, upstream river, and the geometrical shape of branches has been carried out by a few researchers. For example, Buschman et al. [2], using numerical model analysis, pointed out that the inequality of subtidal flow division is affected by the geometrical shape of the channel such as depth,

length, bed roughness, and river discharge. They also emphasized that the flow division of discharge at a junction cannot merely be estimated from the ratio of the wetted cross-sectional areas of the two branches, because the distribution of flow is also affected by spring and neap tide [2]. Sassi et al. [3], using numerical modeling, highlighted the effect of tide on river flow division, where the inequality increases with the bifurcation order. They also found that during the neap tides, the flow may enter the other branch, leading to an unequal discharge distribution. Zhang et al. [4] found that in general, the tides can modify the river discharge distribution over distributaries in the Yangtze estuary. In their numerical result, they also underline that the fortnightly tidal amplitude also contributes to the inequality of subtidal flow division. Moreover, his findings showed that the effect of tidal range on the inequality of flow division is significant. It is important to note that the previous researchers mentioned above focus on subtidal flow division.

However, previous studies mentioned above do not consider the phase difference between the tidal discharge and water level at a tidal channel junction. Indeed, investigating the behavior of phase difference in estuaries still receives little attention, specifically at a junction of estuaries. Horrevoets et al. [5] investigated the phase difference along the single channel of the estuary and suggested that river discharge can control the phase difference. They pointed out that the phase difference in the downstream area is mostly constant, whereas in the upstream part of estuary, the phase difference is not constant. Additionally, the phase difference can become negative in the upstream area. Savenije et al. [6] stated that the phase difference is a basic and significant parameter to describe the tidal wave propagation in an estuary. It is strongly affected by estuary shape and is also a function of the ratio between bank convergence and tidal wave length.

Nevertheless, based on the previous works mentioned above, the temporal variation of phase difference at a tidal channel junction is still unidentified. The change of phase difference in an estuarine system can identify the hydrodynamic processes such as amplification and damping of tidal wave [5,6]. Besides, when the phase difference is near quadrature (close to ~90°), the duration asymmetry of water level can induce asymmetries in tidal current magnitude in estuary channel [7]. More importantly, the phase difference can even influence subtidal transport of flow, sediment, and saltwater in the estuarine system [8,9].

Herein, the chief goal of this work is to shed light on the hydrodynamic aspects of a tidal channel junction monitored by means of an advanced hydro-acoustic system with high-frequency resolution, focusing on the following: (i) the temporal variation of flow division between two seaward branches that are connected to the tidal channel junction, and (ii) identifying the phase difference in tidal discharge in the channels that are connected to the junction.

To further explore the phase difference at a junction, two unidentified issues have been taken into account: (i) the temporal variation of the phase difference in each branch connected to the junction, and the dominant factors that influences the phase difference between the two seaward branches; (ii) the influence of spring-neap tide on the phase difference.

The structure of the conducted study is as follows. Section 2 introduces the field site, the overview of the fluvial acoustic tomography system (FATS), acquisition of FATS data, acoustic Doppler current profiler (ADCP) measurements, bathymetry survey, index-velocity method, and the wavelet analysis. Section 3 presents the results comprising the bathymetry, index velocity relation, tidal discharge result at the junction, discharge validation, streamflow division analysis between the eastern and western branches, and phase difference at the tidal channel junction. Section 4 presents the discussion. The last section outlines the main conclusions.

## 2. Materials and Methods

### 2.1. Field Site

The Ota River estuary is a small-scale estuary with a multi-channel network that connects to the Hiroshima Bay, located in Hiroshima City, Japan. Historically, the channel network of Ota River was

formed on the Ota River delta [10]. The Ota River has important environmental qualities, and the downstream area is also known as a habitat of many creatures, particularly oysters.

As shown in Figure 1a, the Ota river has an apex junction with an asymmetric branch pattern and a north–south oriented branch. It consists of the Ota River floodway (the western-most branch), Tenma River, Kyu Ota River, and the Motoyasu and Enko Rivers (the eastern-most branch) as the five main branches.

The channel network of the Ota River estuary is characterized as a mixed-semidiurnal and mesotidal system with a tidal range varying from 2 to 4 m. The catchment area of the Ota River is approximately 1710 km$^2$. The freshwater discharge that is monitored at the Yaguchi gauging station mostly varies between 20 to 50 m$^3$s$^{-1}$ during the low-flow condition, except during periods of heavy rainfall.

There are two sluice gates at the Ota River floodway positioned near the first bifurcation channel (see Figure 1a). The sluice gates of Gion consist of a movable weir with three gates that only open 10% during normal operation; the Oshiba gates consist of a movable weir with three gates that are completely open throughout the year.

The studied junction is in the middle part of the Ota River network channel, approximately 2.5 km downstream from the main bifurcation channel (apex junction) and approximately 5.8 km from the river mouth, as shown in Figure 1b. The tidal channel junction consists of three branches: the northern branch (upstream Kyu Ota River), the eastern branch (downstream Kyu Ota River), and the western branch (Tenma River). As an estuary system with multi-channel network, the saltwater intrusion in the Ota River can reach 11.5 km upstream from the river mouth. The bed materials of the channel network mostly consist of sand containing less silt and clay.

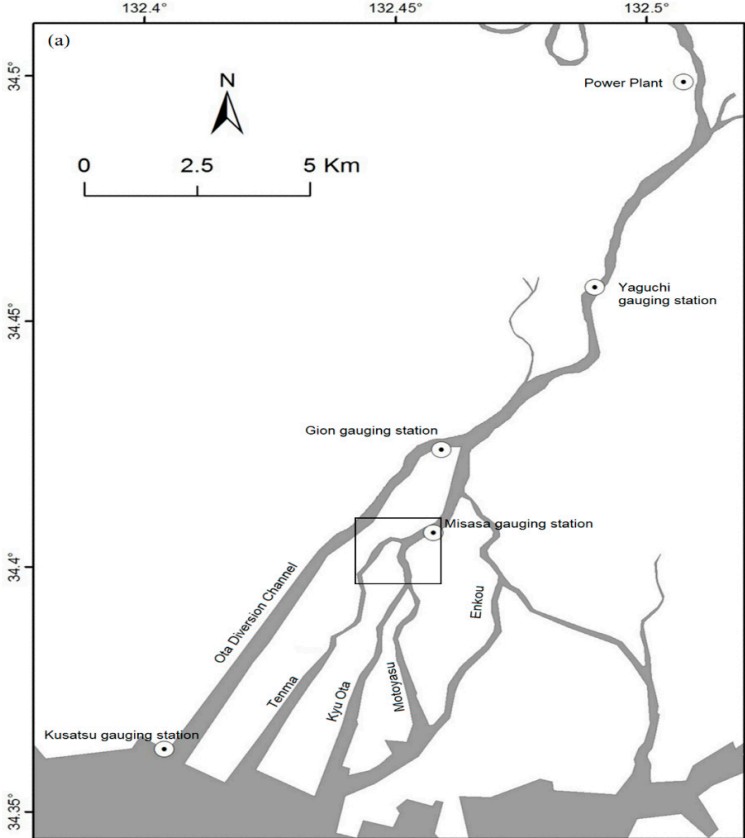

**Figure 1.** *Cont.*

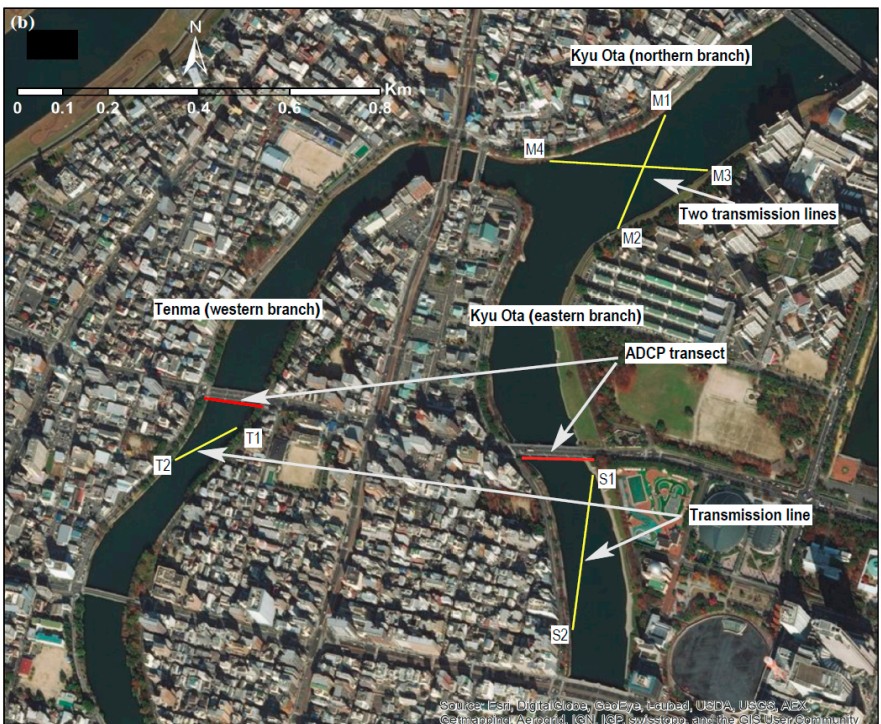

**Figure 1.** (**a**) Ota river channel network map. The black box denotes the study area. (**b**) Observation site and deployment of the fluvial acoustic tomography system (FATS) at the eastern, western, and northern branches. The yellow lines represent the transmission lines between the two transducers of FATS. S1–S2, T1–T2, and M1–M4 denote the locations of the transducers for FATS measurements. The red lines denote the acoustic Doppler current profiler (ADCP) transects for the comparison with FATS.

## 2.2. Measurement Method

### 2.2.1. The Overview of the Fluvial Acoustic Tomography System

The FATS developed at Hiroshima University was used to continuously measure the time series of tidal discharge at the tidal channel junction. The advantages of FATS are that it enables the measurement of discharge continuously in the long-term period that is suitable to be used even in such a tidal river with salt water intrusion, extremely shallow water depth area, and a larger width-to-depth ratio. The sound rays of FATS are capable of covering the whole cross-section of the stream and capturing the depth- and range-averaged velocity. As a result, the FATS can accurately estimate the cross-sectional averaged velocity [11]. To obtain the discharge ($Q$), the cross-sectional averaged velocity is multiplied by cross-sectional area ($A_0$), which varies in time and corresponds to water level fluctuation. For detailed information about the FATS measurement method and its reliability, readers can refer to previous studies [11–14].

### 2.2.2. Acquisition of FATS Data

Continuous measurements of discharge by the FATS were carried out from 8 to 20 June 2017 and 9 to 29 June 2017 for the western and eastern branches, respectively (see Table 1). Two pairs of FATS were deployed at the western and eastern branches located near a tidal channel junction, as shown in Figure 1b. The acoustic pulses (central frequency: 30 and 53 kHz) were transmitted concurrently from both transducers every 30 s. The transducers were installed at the height of 0.2 m above the channel bed using stands.

**Table 1.** Summary of the fluvial acoustic tomography system (FATS) experiment period.

| Date of FATS Experiment | Deployment Site | Method of Discharge Estimation |
| --- | --- | --- |
| 9 to 29 June 2017 | Eastern branch | Index-velocity method as described by Kawanisi et al. [11] |
| 8 to 20 June 2017 | Western branch | Index-velocity method as described by Kawanisi et al. [11] |
| 6 to 29 November 2017 | Northern branch | Two-crossing transmission lines as described by Kawanisi et al. [13] and Bahreinimotlagh et al. [14] |

On the other hand, the monitoring of tidal discharge at the northern branch was conducted from 6 to 29 November 2017 to obtain additional data related to the phase difference investigation at the junction. In this case, we use two pairs of FATS with a two-crossing transmission lines configuration to estimate tidal discharge, as proposed by Kawanisi et al. [13] and Bahreinimotlagh et al. [14]. This method is applied to show the significant variation that occurs during field investigations.

The coordinates of the transducers and the distance between transducers for the western, eastern, and northern branches are presented in Table 2.

**Table 2.** Coordinates of transducers.

| Code | River branch | Transducers | Latitude (°) | Longitude (°) | The Distance between Transducers (m) |
| --- | --- | --- | --- | --- | --- |
| Eastern branch | Kyu Ota river | $S_1$ | 34°24′00.60″ | 132°27′08.64″ | 246.363 |
|  |  | $S_2$ | 34°23′52.68″ | 132°27′07.32″ |  |
| Western branch | Tenma river | $T_1$ | 34°24′03.80″ | 132°26′43.63″ | 158.836 |
|  |  | $T_2$ | 34°24′01.09″ | 132°26′38.34″ |  |
| Northern branch | Kyu Ota river | $M_1$ | 34°24′20.36″ | 132°27′18.00″ | 289.744 |
|  |  | $M_2$ | 34°24′20.09″ | 132°27′06.66″ |  |
|  |  | $M_3$ | 34°24′23.49″ | 132°27′14.52″ | 224.639 |
|  |  | $M_4$ | 34°24′16.68″ | 132°27′11.38″ |  |

### 2.2.3. Index-Velocity Method

Application of the index velocity method for calculating continuous records of discharge has become increasingly common, particularly since the development of acoustic-based instruments such as acoustic velocity meters (AVMs) and horizontal acoustic Doppler current profilers (H-ADCPs) and acoustic tomography systems [11,15–17]. In this study, the index velocity method (IVM) is used to estimate the discharge based on the regression equation obtained from the relationship between the velocity parameter of ADCP measurement and velocity along the transmission line of FATS measurement, as described in the previous work of Kawanisi et al. [11]. Subsequently, the discharge can be performed as a product of the regression equation and cross-sectional area, which is a function of water depth.

### 2.2.4. ADCP Measurements and Cross-Sectional Area Determination

In this study, Teledyne RDI StreamPro ADCP was used to provide and establish reference discharge data for validating FATS measurements between two seaward branches. Each ADCP campaign was conducted approximately every 5–8 min to collect the data at the bridge near the transducers (see Figure 1b). However, for the validation of FATS measurement in the northern branch, the ADCP campaign was carried out along the transmission line between two transducers. StreamPro ADCP was set to operate in water mode 12 with a bin size of 12 cm, the number of bin sizes set to 30, with 6 pings per ensemble. In this experiment, ADCP discharges can be obtained using WinRiver II software, in which each of the bad bin ensemble parts did not exceed 2%, while bad ensembles did not exceed 3% in each transect.

To obtain the cross-sectional area ($Ao$) as a function of water level, water level data from three branches were recorded every 10 min using water level loggers (Hobo U20-001-01-Ti, Onset Co.) that were attached to the transducer stands. To obtain water level estimates, the measured pressure values

should be normalized with respect to atmospheric pressure, as recorded by the barometer deployed on the riverbank. The accuracy of the water level sensors and the barometer was ±5 and ±3 mm $H_2O$, respectively. The post-processing of the water pressure and barometric data as parameter inputs to the software program Hoboware Pro [18].

### 2.2.5. Bathymetry Survey

The bathymetric survey of the three branches was conducted using an autonomous boat equipped with global positioning system (GPS) device (resolution 1/10000 s) and a single beam echo sounder at a frequency of 200 kHz (resolution: 0.01 m). Transect lines were identified according to the site accessibility conditions that were assessed in the field inspection. Thus, in the case of our monitoring program, we performed bathymetry transects for 53 sections, as depicted in Figure 2a. The directions of these lines are perpendicular to the stream direction, with the distance of each line being approximately 50 m (i.e., the distance between the left and right banks ≈50 m). Each transected line represents two points with the north–east coordinates based on the Japanese datum standard system. Output data of bathymetry transects obtained by the boat were stored as x, y, and z positions, where x and y represent the horizontal positions obtained from GPS recording, and z was the water depth measured by the echo sounder. The survey started at the upstream junction and finished at the downstream junction of both branches.

### 2.2.6. Wavelet Analysis

Wavelet transformation is an advanced analysis method in signal processing particularly used in investigating hydrodynamic processes in an estuary such as a tidal wave and its interaction with the river discharge [3,8,19]. In this study, three functions of wavelet were used to analyze the interaction between tidal discharge and water level, that is, continuous wavelet transformation (CWT), cross-wavelet transformation (XWT), and wavelet coherence (WTC).

CWT is used to detect variations in time series and their simultaneous representation in the time–frequency space, and also to determine the dominant period. Moreover, CWT analysis can display temporal and spatial evolutions of tidal frequency spectra [19]. XWT was utilized to analyze the phase difference between two-time series. The phase difference is basically the phase angle represented by the arrow, that is, the arrow pointing right means both signals travel in the same direction (in-phase), the arrow pointing left means both signals travel in the opposite direction (anti-phase), and the arrow pointing down means the first time series of signal leads the second one by 90°. WTC is used to find significant coherence between the two-time series and to show the confidence level. Details of wavelet are referred to in Grinsted et al. [20].

## 3. Results

### 3.1. Bathymetry at the Tidal Channel Junction

Figure 2a shows the bathymetry in three branches at a tidal channel junction where the bed level varies between −1.5 to −4.5 T.P. m, but mostly dominated by −3 T.P. m and −2 T.P. m for the eastern and western branches, respectively. From Figure 2a,b, it is revealed that the eastern branch is deeper and wider than the western branch. Moreover, sedimentation was found in the western branch around 800 m from transect 1. Similarly, the bathymetry in the northern branch shows that the width of the northern branch is wider compared with those of the eastern and western branches. Moreover, the bed level of the northern branch is relatively constant from −2.5 T.P. m to −3.0 T.P. m.

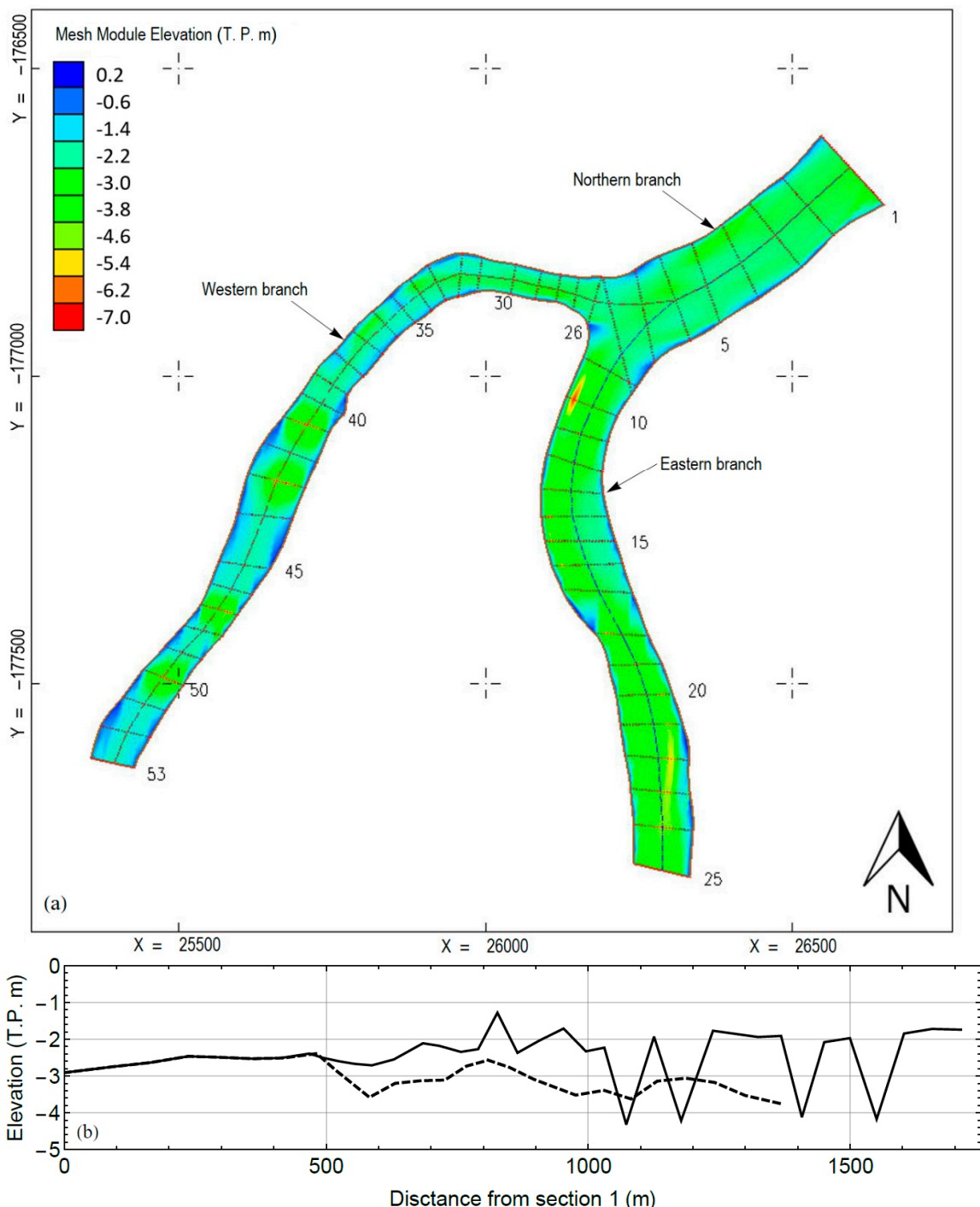

**Figure 2.** (**a**) Bathymetry map around the bifurcation. The elevation is measured as a vertical distance (T.P. m). The listed numbers in the range of 1–53 denote the transects of the cross-sectional line measured by the Coden boat RC–S3. (**b**) The longitudinal distributions of mean bed level. The black line and the black dashed lines represent the mean bed levels from the northern to the western and eastern branches, respectively.

### 3.2. Establishing an Index Velocity Rating and Validation of Discharge Measurement Obtained by FATS

We calculated tidal discharge by regressing the FATS index velocity with the velocity of ADCP, yielding discharge after multiplying it with the oblique cross-sectional area ($A_0$) along the transmission line, which varies in time as a function of water depth. Figure 3a,b show the linear regression relationships between the velocity of ADCP ($Q_{ADCP}/A_0$) and the FATS velocity along the transmission

line ($u_m$) for the eastern and western branches, respectively. Both regressions show a high correlation with the $R^2 \sim 0.99$, though the data used for validation, particularly at the eastern branch, are limited. However, it is necessary to emphasize that having an additional number of ADCP transects in the eastern branch location is very difficult because of the high number of activities that take place along that branch (e.g., water taxi, fishing), which constrained our validation endeavors.

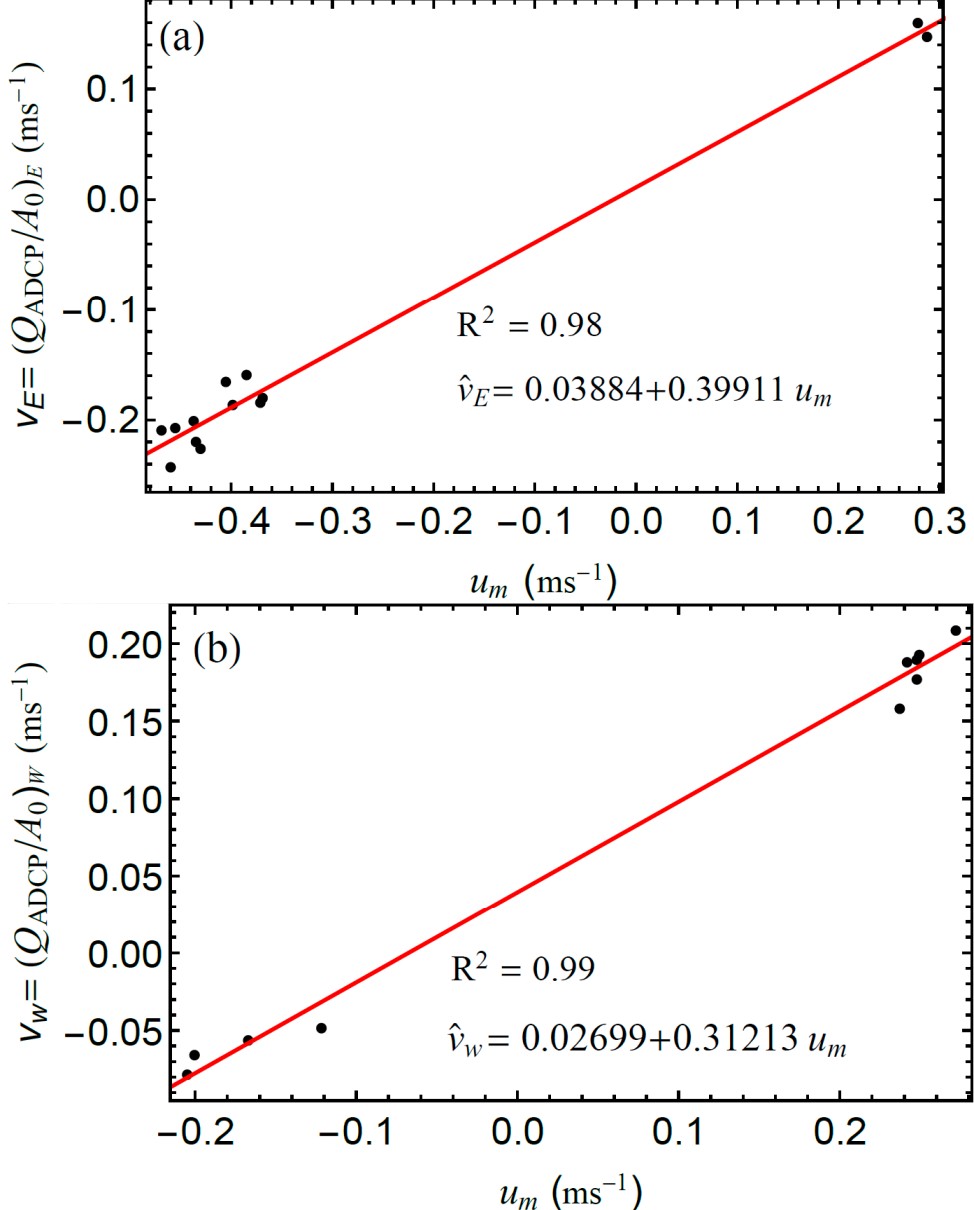

**Figure 3.** Index velocity relations for the (**a**) eastern and (**b**) western branches.

The index-velocity equations used for the calculation of the cross-sectional average velocity at eastern branch ($\hat{v}_E$) and the western branch ($\hat{v}_W$) are as follows:

$$\hat{v}_E = 0.03884 + 0.39911\, u_m, \tag{1}$$

$$\hat{v}_W = 0.02699 + 0.31213\, u_m. \tag{2}$$

Thus, the discharge of FATS for the western and eastern branches can be respectively computed as follows:

$$Q_E = \hat{v}_E \times A_0, \tag{3}$$

$$Q_W = \hat{v}_W \times A_0. \tag{4}$$

The FATS estimates were validated by the moving-boat ADCP measurements, as shown in Figure 4a–c for the eastern, western, and northern branches, respectively. The time series of tidal discharge for the eastern and western branches were obtained from Equations (3) and (4), respectively, whereas the discharge for northern branch was obtained from the two-cross paths of the FATS method adopted from Kawanisi et al. [13] and Bahreinimotlagh et al. [14].

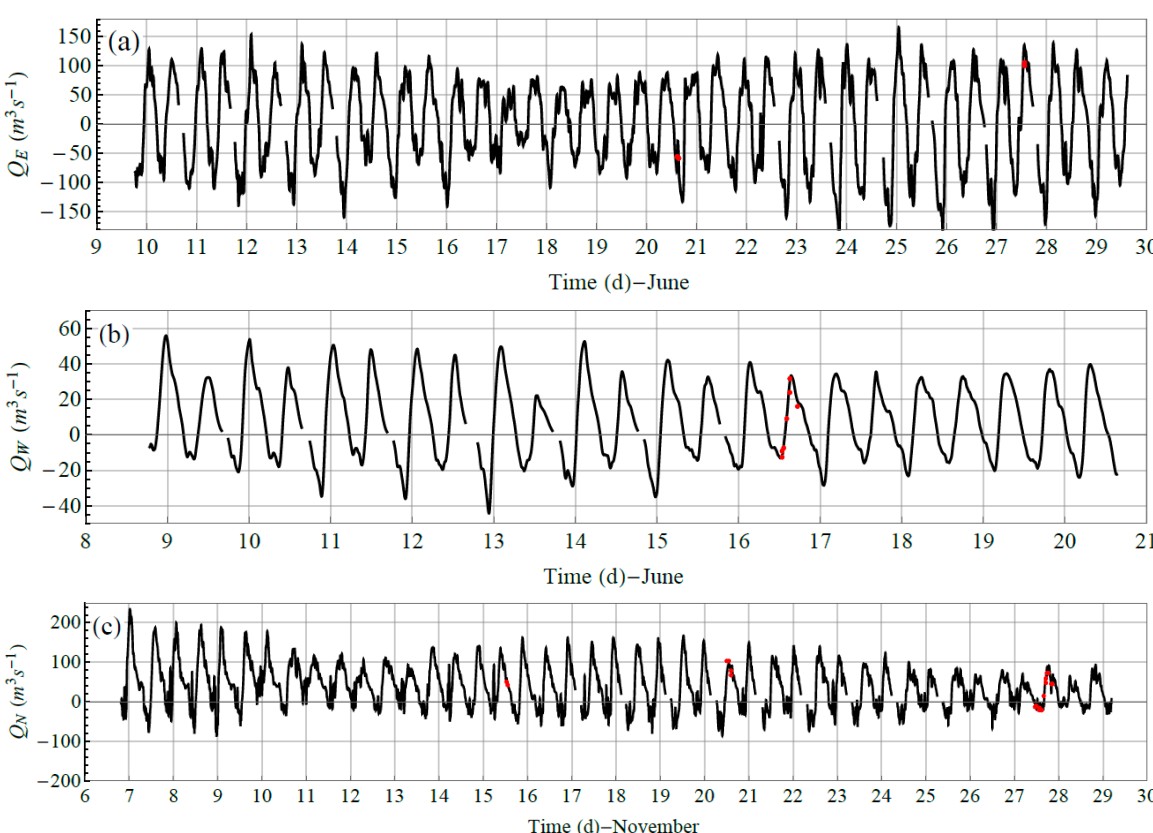

**Figure 4.** Temporal variations of (**a**) tidal discharge at the eastern branch, (**b**) tidal discharge at the western branch, and (**c**) tidal discharge at the northern branch. The red dots denote the discharge obtained using moving-boat ADCP measurements. Discontinuations of the discharge time series correspond to the missing period, as the result of the transducers were not covered by water during low tide.

It is important to note that particularly in the eastern branch, the validation using ADCP measurement is limited as a result of ship traffic. The relative difference of discharges estimated from ADCP measurements and FATS are mostly ranged from ~0.4% to ~10%, as shown in Table 3, Table 4, and Table 5. The results are consistent with those of previous works, where the relative differences between FATS and ADCP estimates range from 1% to 10% [12–14]. Moreover, we can evaluate how closely the FATS estimates match the ADCP results using the equation of root mean square error (RMSE):

$$\text{RMSE} = \sqrt{\frac{\sum_{i=1}^{n}[Q_{\text{FATS}}(t_i) - Q_{\text{ADCP}}(t_i)]^2}{n}}, \tag{5}$$

where $Q_{ADCP}(t_i)$ and $Q_{FATS}(t_i)$ correspond to the observed and calculated discharge at time $t_i$, respectively; and $n$ is the number of data points. Thus, based on the data obtained from Table 3 to Table 5, the root means square error (RMSE) at the eastern branch is 5.0 m³s⁻¹ or only ~1.67% of the discharge range, whereas the RMSE of the western branch is 1.68 m³s⁻¹ or ~1.7% of the discharge range. Similarly, the root means square error (RMSE) at the northern branch is 5.3 m³s⁻¹ or ~1.65% of the discharge range. Thus, FATS measurement results are reliable estimates of the discharge in a tidal estuary.

**Table 3.** Comparison between FATS and acoustic Doppler current profiler (ADCP) measurements at the eastern branch.

| Date | Local Time | $Q_{FATS}$ (m³s⁻¹) | $Q_{ADCP}$ (m³s⁻¹) | $\Delta Q = Q_{FATS} - Q_{ADCP}$ (m³s⁻¹) | $\Delta Q / Q_{ADCP}$ (%) |
|---|---|---|---|---|---|
| June 20, 2017 | 14:44:05 | −57.56 | −55.67 | −1.89 | 3.40 |
| June 20, 2017 | 14:55:13 | −58.53 | −55.67 | −2.86 | 5.14 |
| June 20, 2017 | 15:07:31 | −59.09 | −58.81 | −0.28 | 0.48 |
| June 20, 2017 | 15:16:31 | −60.31 | −58.41 | −1.90 | 3.25 |
| June 20, 2017 | 15:27:11 | −65.81 | −58.18 | −7.63 | 13.11 |
| June 27, 2017 | 13:29:25 | 108.38 | 99.55 | 8.83 | 8.87 |
| June 27, 2017 | 13:41:38 | 109.60 | 104.65 | 4.95 | 4.73 |

**Table 4.** Comparison between FATS and ADCP measurements at the western branch.

| Date | Local Time | $Q_{FATS}$ (m³s⁻¹) | $Q_{ADCP}$ (m³s⁻¹) | $\Delta Q = Q_{FATS} - Q_{ADCP}$ (m³s⁻¹) | $\Delta Q / Q_{ADCP}$ (%) |
|---|---|---|---|---|---|
| June 16, 2017 | 12:45:11 | −11.75 | −12.2 | 0.45 | -3.69 |
| June 16, 2017 | 12:59:19 | −10.26 | −9.13 | −1.13 | 12.38 |
| June 16, 2017 | 13:09:38 | −8.05 | −7.37 | −0.68 | 9.23 |
| June 16, 2017 | 13:14:28 | 9.80 | 8.91 | 0.89 | 9.99 |
| June 16, 2017 | 14:14:28 | 25.19 | 24.26 | 0.93 | 3.83 |
| June 16, 2017 | 14:53:59 | 31.85 | 31.47 | 0.38 | 1.21 |
| June 16, 2017 | 15:13:13 | 31.17 | 27.17 | 4.00 | 14.72 |

**Table 5.** Comparison between FATS and ADCP measurements at the northern branch.

| Date | Local Time | $Q_{FATS}$ (m³s⁻¹) | $Q_{ADCP}$ (m³s⁻¹) | $\Delta Q = Q_{FATS} - Q_{ADCP}$ (m³s⁻¹) | $\Delta Q / Q_{ADCP}$ (%) |
|---|---|---|---|---|---|
| November 15, 2017 | 12:46:48 | 52.98 | 49.68 | 3.3 | 6.64 |
| November 15, 2017 | 13:22:43 | 43.67 | 42.95 | 0.72 | 1.68 |
| November 20, 2017 | 12:15:03 | 88.61 | 103.34 | −14.73 | −14.25 |
| November 20, 2017 | 13:14:28 | 95.68 | 104.48 | −8.80 | −8.42 |
| November 20, 2017 | 14:06:36 | 78.95 | 78.64 | 0.31 | 0.39 |
| November 27, 2017 | 11:37:19 | −10.07 | −10.75 | 0.68 | −6.33 |
| November 27, 2017 | 13:13:52 | −17.80 | −19.41 | 1.61 | −8.29 |
| November 27, 2017 | 14:34:03 | −18.16 | −20.15 | 1.99 | −9.88 |
| November 27, 2017 | 16:09:35 | 14.32 | 15.24 | −0.92 | −6.04 |
| November 27, 2017 | 16:57:07 | 57.83 | 57.73 | 0.10 | 0.17 |
| November 27, 2017 | 17:35:10 | 73.36 | 73.61 | 3.75 | 5.09 |
| November 27, 2017 | 18:20:39 | 77.10 | 71.55 | 5.55 | 7.76 |
| November 27, 2017 | 19:49:15 | 41.15 | 45.65 | −4.50 | −9.86 |
| November 27, 2017 | 20:24:41 | 48.85 | 44.98 | 3.87 | 8.60 |

*3.3. Time Series of Tidal Discharge in the Western, Eastern, and Northern Branches*

Figure 5a shows the temporal variations of the water levels and discharges in the eastern and western branch. The time series of tidal discharge for the eastern and western branches were obtained from Equations (3) and (4), respectively. The tidal discharge shows the semidiurnal characteristic, that

is, the high tide and low tide occur two times a day. The discharges in the branches show distinct characteristics, where the eastern and western branches range from −157 to 155 m³s⁻¹ and from −42 to 58 m³s⁻¹, respectively. From the comparison of the tidal discharges at the eastern and western branches, as shown in Figure 5a, it is revealed the tidal discharge of the eastern branch is approximately two to six times greater than that of the western branch during flood and ebb tides. This is caused by the difference in the geometry of both branches, that is, the water depth and the channel width of the eastern branch are much larger than that of the western branch, as can be seen in Figure 2. Moreover, the pattern of tidal discharge at the western branch is also much more asymmetric compared with that of the eastern branch, where the discharge during the ebb tide was higher than during the flood tide.

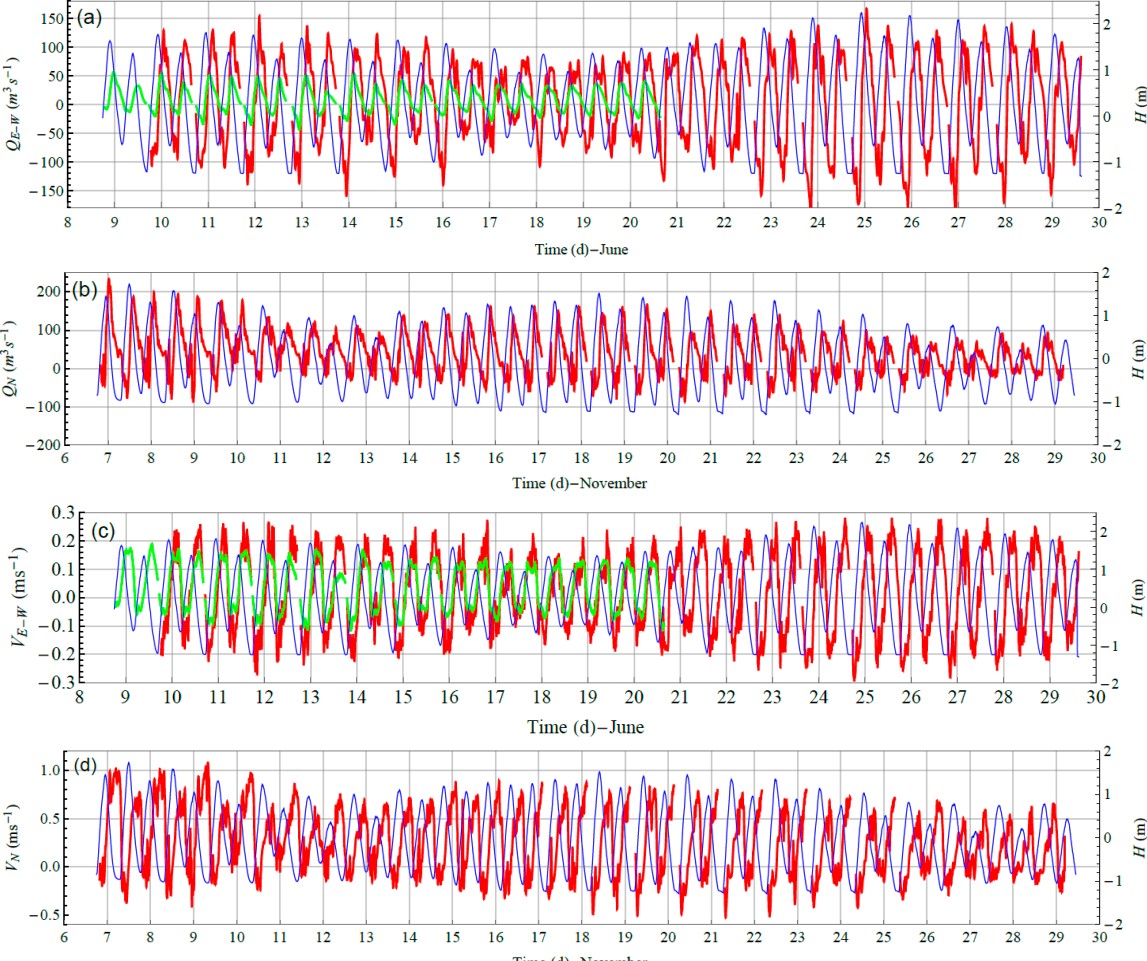

**Figure 5.** Temporal variations of (**a**) water level at the eastern branch (blue line), tidal discharge at the eastern branch (red line), and tidal discharge at the western branch (green line). (**b**) Water level at the northern branch (blue line) and tidal discharge at the northern branch (red line). (**c**) Water level at the eastern branch (blue line), tidal velocity at the eastern branch (red line), and tidal velocity at the western branch (green line). (**d**) Water level at the northern branch (blue line) and tidal velocity at the northern branch (red line). Positive discharge and velocity coincide with the seaward flow.

Similarly, Figure 5b shows the temporal variation of water level and tidal discharge at the northern branch located at the upstream junction, where the tidal discharge ranged from −90 to 230 m³s⁻¹. In addition, Figure 5c shows the temporal variation of the water level and tidal velocity for the eastern and western branch, whereas Figure 5d describes the tidal velocity and water level at the northern branch. The temporal variation of tidal velocity at the eastern, western, and northern branch ranges from −0.28 to 0.3 ms⁻¹, −0.12 to 0.19 ms⁻¹, and −0.5 to 1 ms⁻¹, respectively.

### 3.4. Subtidal Discharge Division between the Western and Eastern Branches

The purpose of quantifying subtidal discharge division is to explore the variation of flow division and the inequality of streamflow during the spring and neap tide. In this study, we use the subtidal discharge obtained by applying a Battle–Lemarie filter of wavelet function. The division of subtidal discharge between the eastern and western branches at a tidal channel junction can be quantified using the discharge asymmetry index ($\psi$), as proposed by Buschman et al. [2].

$$\psi = \frac{\langle Q_E \rangle - \langle Q_W \rangle}{\langle Q_E \rangle + \langle Q_W \rangle} \qquad (6)$$

The tidal discharge asymmetry index ($\psi$) is zero for an equal discharge division, $\psi = 1$ when all river water flows through the eastern branch, and $\psi = -1$ when the western branch carries all the discharge. Therefore, a positive value is obtained when the discharge at the eastern branch is greater than that at the western branch, and vice versa.

Only data from 10 and 21 June, 2017 were used to calculate the discharge asymmetry index ($\psi$) using Equation (6), because of the difference in lengths of the data obtained from the FATS measurements between the eastern and western branches. Figure 6 shows the discharge asymmetry index and subtidal discharge variation in the eastern and western branch from 10 to 21 June, suggesting that the temporal variations in subtidal discharge of the eastern branch are slightly larger than those of the western branch, except within 18 and 19 June, the variation was almost the same during the neap tide. Figure 6, which captures a spring-neap fluctuation, shows the temporal variation in the discharge asymmetry index. The discharge asymmetry index fluctuates in the range of −0.02 to 0.26 from 10 to 21 June, indicating the inequality of flow division between the eastern and western branch, where the eastern branch can distribute larger discharge than that in the western branch during spring tide; the flow division between the western and eastern branch is nearly equal during the neap tide (18 to 19 June).

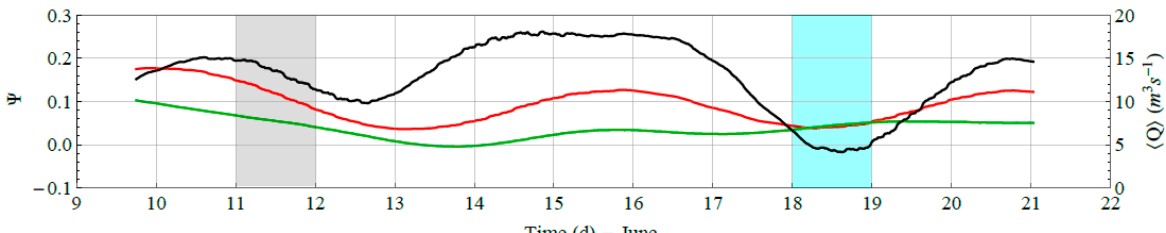

**Figure 6.** Discharge asymmetry index (black line). Temporal variations of subtidal discharges in the eastern (red line) and western branch (green line). The grey and blue highlighted boxes represent the spring and neap tide, respectively.

### 3.5. The Wavelet Analysis for the Interaction between the Water Level and Tidal Discharge in the Three Branches

In this section, three functions of wavelet analysis are used to investigate the time-series signal between tidal discharge and water level in three branches connected to a channel junction. In this study, XWT is used to determine the interaction between tidal discharge and water level, that is, propagating, standing wave, and mixed wave. Tidal discharge ($Q$) varies along the channel in estuaries and co-varies with the tidal wave. There are two methods to determine the type of wave in estuarine channels, that is, phase lag and phase difference [21,22]. The phase lag is the time lag between high water slack (HWS, i.e., when the discharge is zero) and high water (HW, i.e., when the water level is maximum) [8,21]. In contrast, the phase difference is the time lag between the peak tidal discharge and the HW [21,22]. Therefore, there is a relationship between the phase difference and the phase lag, that is, $\varepsilon + \varphi = \frac{\pi}{2}$, where $\varepsilon$ is the phase lag and $\varphi$ is the phase difference [8,23]. In this study, we use phase

difference to describe the relationship between tidal discharge and water level [22,24]. If φ = 90° is a standing wave, φ = 0° is propagating, and 0° < φ < 90° is a mixed wave.

Figure 7 shows three figures consisting of CWT, XWT, and WTC analyses between the tidal discharge and water level at the eastern branches. From the figures, the dark red color background denotes a strong power spectrum, suggesting a dominant tidal signal. The vertical bar shows the energy of the tidal domain. From the CWT and WTC analyses, the semidiurnal signal is more dominant than the diurnal signal. Moreover, the semidiurnal domain is always present during the spring and neap tide, whereas the diurnal domain is only present during spring tide, but fades during the neap tide. Thus, the tide of the Ota River is characterized by the mixed-semidiurnal tide. Besides the semidiurnal and diurnal signal, we can also identify the quarter-diurnal in the eastern branch, which is discontinuous and weak. In contrast, the fortnightly signal cannot be detected. From XWT analysis, the phase difference between the tidal discharge and the water level at the eastern branch can be denoted by the black arrows pointing straight down at ~78°.

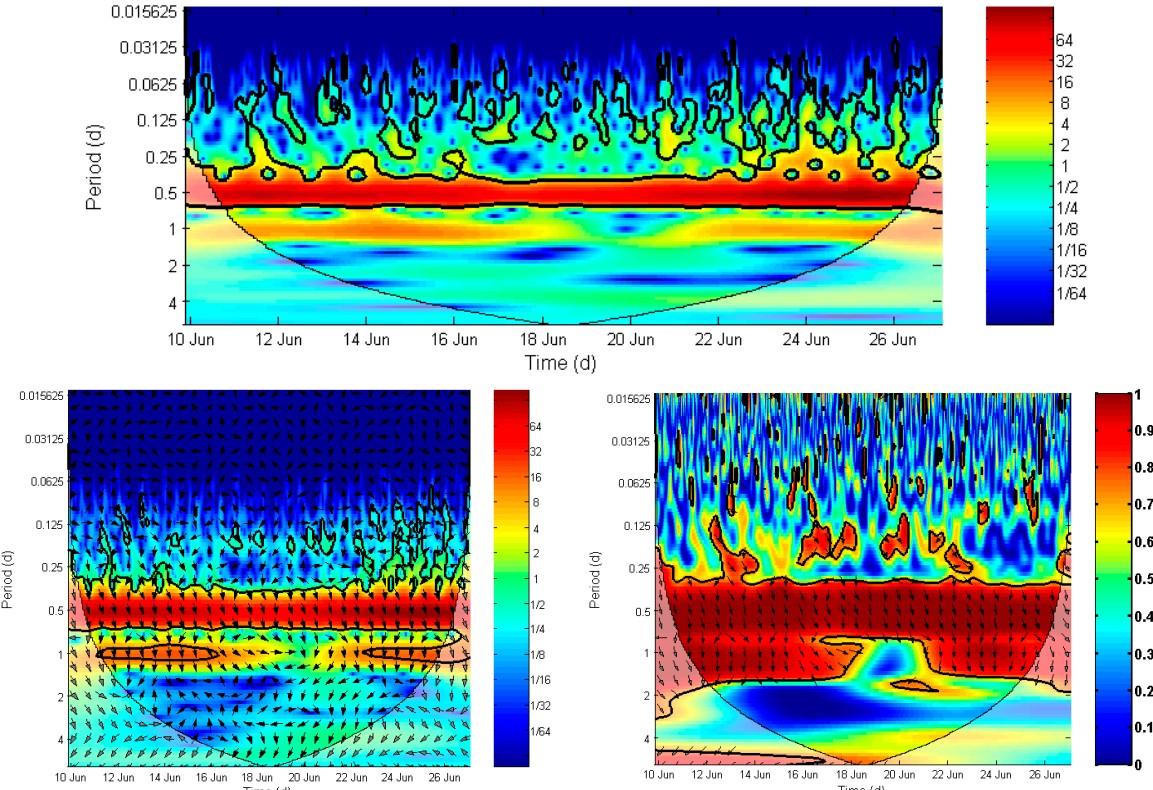

**Figure 7.** Wavelet analyses in eastern branch: (top) continuous wavelet transformation; (bottom left) cross-wavelet transform; (bottom right) wavelet coherence. The black contours represent the 0.95 confidence level against red noise, and the cone of influence (COI) where edge effects might distort the picture is shown as a lighter shade. The relative phase relationship is shown as arrows (with in-phase pointing right, anti-phase pointing left, and the first time series leading the second one by 90° pointing down). The wavelet power is $\log_2(A^2/v)$, where $A$ is the wavelet amplitude and $v$ is the variance of the original tidal signals. The $y$-axis in the CWT and XWT is on a $\log_2$ scale.

Similarly, Figure 8 shows three plots consisting of CWT, XWT and WTC analyses between the tidal discharge and water level at the western branches. The semidiurnal domain is always present, whereas the diurnal domain is only present during spring tide but diminishes during the neap tide. We can recognize the quarter-diurnal in the western branch which is discontinuous and weak, but more pronounced compared to the eastern branch. The fortnightly signal is also cannot be identified in the western branch.

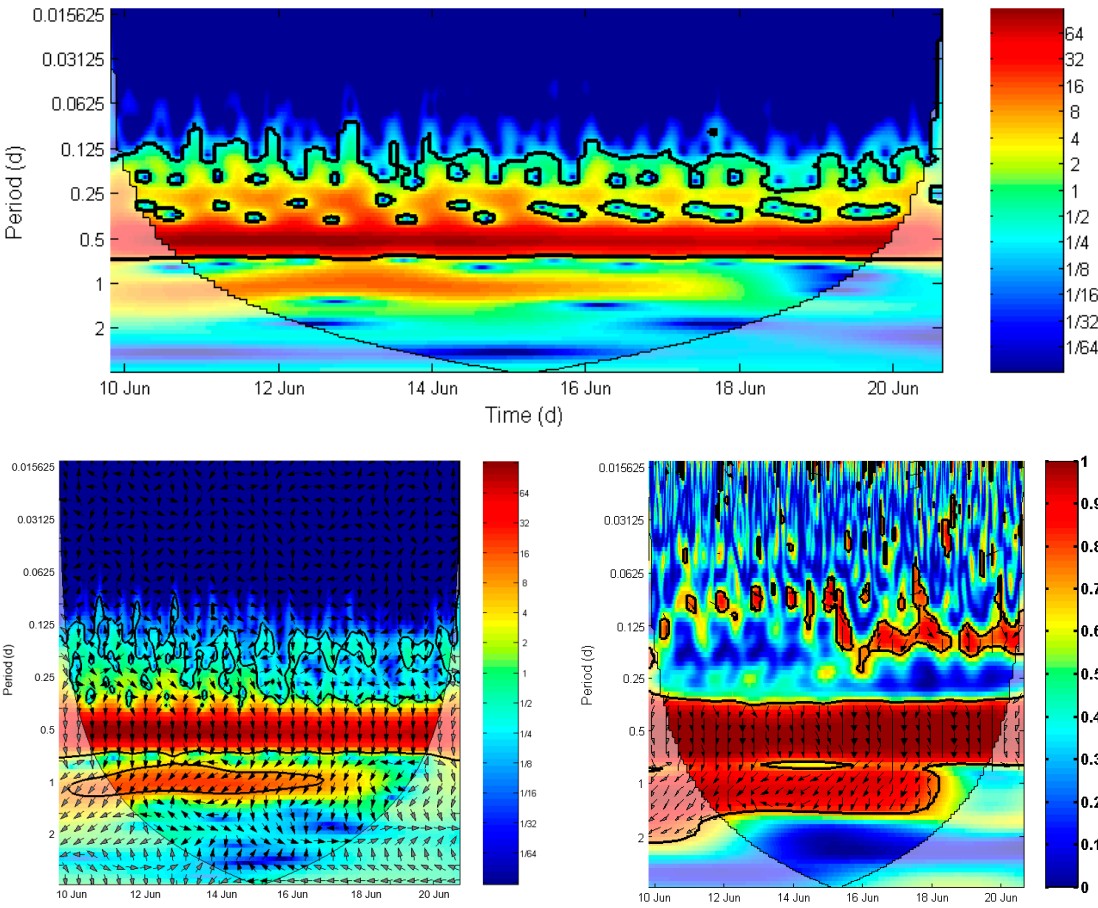

**Figure 8.** Wavelet analyses in western branch: (top) continuous wavelet transform. (bottom left) cross-wavelet transform. (bottom right) wavelet coherence. The black contours represent the 0.95 confidence level against red noise and the cone of influence (COI) where edge effects might distort the picture is shown as a lighter shade. The relative phase relationship is shown as arrows (with in-phase pointing right, anti-phase pointing left, and the first time series leads the second one by 90° pointing down). The wavelet power is $\log_2(A^2/v)$, where $A$ is the wavelet amplitude, $v$ is the variance of the original tidal signals. The *y*-axis in the continuous wavelet transformation (CWT) and cross-wavelet transformation (XWT) is on a $\log_2$ scale.

From XWT analysis, the phase difference between the tidal discharge and the water level at the western branch shows that the black arrows are pointing down at ~88.5°.

The phase difference in tidal discharge between the western and eastern branches was further examined to ensure that the tidal discharge phase of the eastern branch is greater than that of the western branch. Figure 9 confirms that the tidal discharge phase at the western branch is larger than the discharge phase at the eastern branch, with a phase difference of ~11°. Moreover, it also seems that the geometry of the branch has a greater effect on the behavior of the phase difference at the western branch compared with the eastern branch.

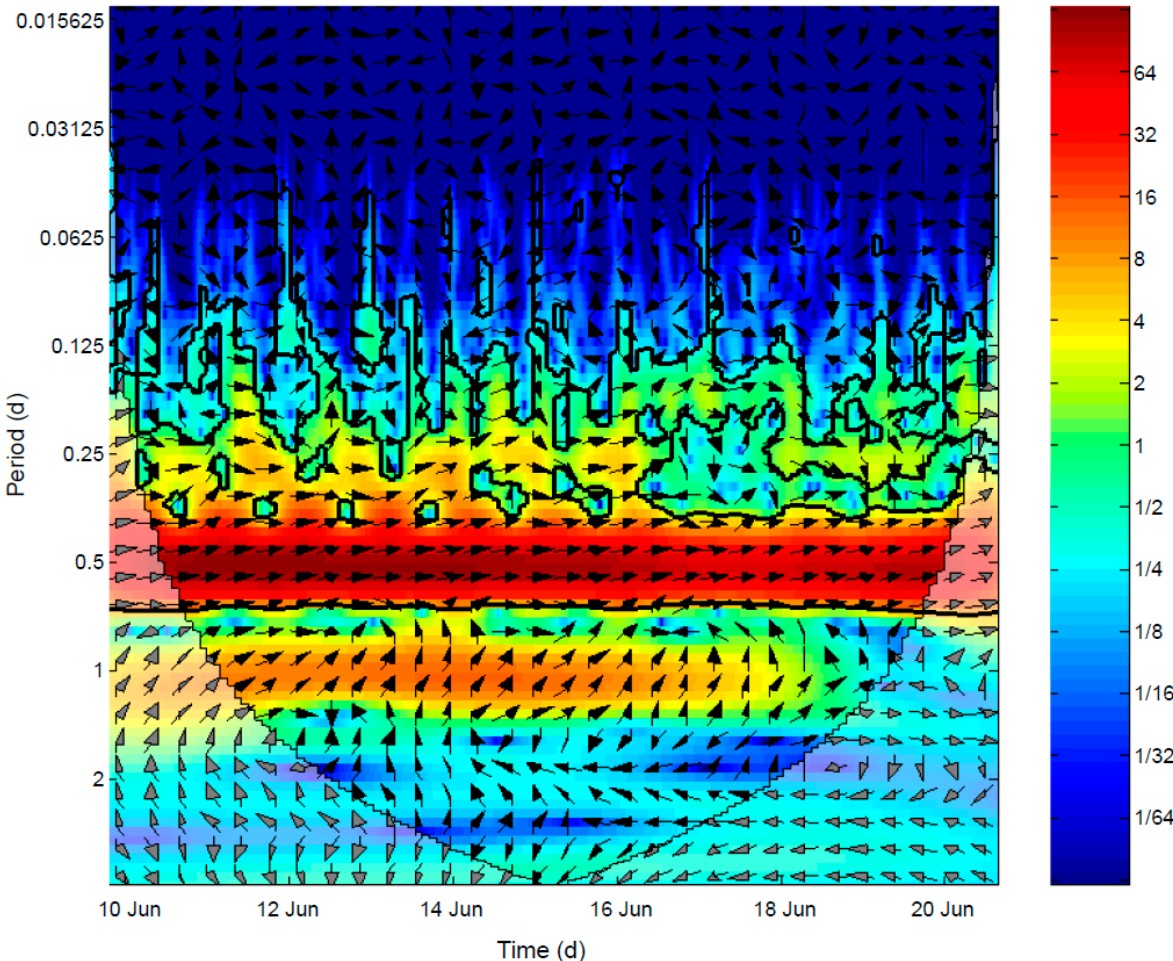

**Figure 9.** The phase difference in the tidal discharge phase between the eastern and western branches. The black contours represent the 95% confidence level for red noise, and the cone of influence (COI) where edge effects might distort the picture is shown as a lighter shade. The relative phase relationship is shown as arrows (with in-phase pointing right, anti-phase pointing left, and the first time series leading the second one by 90° pointing down). The wavelet power is $\log_2(A^2/v)$, where $A$ is the wavelet amplitude and $v$ is the variance of the original tidal signals. The $y$-axis is on a $\log_2$ scale.

As mentioned in Section 2.2.2, to gain a comprehensive understanding of phase difference behavior around the junction, an additional observation campaign at the northern branch using FATS is needed to collect the tidal discharge continuously. With this, the phase difference before and after the junction can be investigated completely (see Figure 1b).

The phase difference during semidiurnal is ~90°. This result confirms that the phase difference at the northern branch shows slightly increased behavior compared with the seaward branches (eastern and western branches). This result indicates that the behavior of phase difference is slightly increased after passing through the junction into the northern branch (a landward branch of the junction).

## 4. Discussion

### 4.1. Interpretation of Flow Division between Two Seaward Branches

From the bathymetry map presented in Figure 2, the depth and width of the eastern branch are greater than those of the western branch. However, the inequality of flow division at the tidal channel junction cannot be calculated using only a ratio of the wetted cross-sectional areas between the two branches [2], because the hydrodynamic process at the tidal channel junction is also strongly

influenced by the spring and neap tide, as well as by the asymmetrical geometry shape between the two seaward branches.

The response of tidal dynamic to the local geometry between two seaward branches shows distinct characteristics, where the asymmetric discharge at the western branch is more pronounced compared with that at the eastern branch. The asymmetric tidal discharge observed at the western branch indicates the ebb discharge is larger than the flood discharge. On the contrary, the characteristic of tidal discharge at the eastern branch is nearly equal during the ebb and flood tide. As we can see in Figure 2, the western branch is shallower and narrower than the eastern branch. Therefore, the propagating tidal wave in shallow and narrow branches, and modulated with the river discharge, can generate the nonlinear effect as a result of interaction with the local topography [9]. This nonlinear effect leads to the generation and development of shallow-water constituents, which cause tidal distortion and asymmetry [25]. This asymmetric tidal discharge can induce a seaward sediment movement [26].

Regarding the flow division characteristic at the tidal channel junction of Ota River, our findings are depicted in Figure 6. The discharge asymmetry index ranges from −0.02 to 0.26 during the 10 to 21 June, suggesting that the eastern branch has the capability to deliver greater amounts of subtidal discharge, approximately 55–63% compared with the western branch. However, the flow division is nearly equal during the neap tide (i.e., 18 to 19 June). Figure 6 shows that the equality of asymmetry index between the eastern and western channels can be observed clearly during the neap tide period. Nevertheless, the inequality of flow division is obviously prominent during the spring tide duration. The fluctuation in the subtidal discharge at the eastern branch is greater than that at the western branch; the subtidal discharge at the eastern branch decreases during the neap tide, so that both subtidal discharges are nearly equal during the neap tide.

### 4.2. Interpretation of the Interaction between the Tidal Discharge and Water Level in the Three Branches

CWT analysis identifies the semidiurnal and diurnal signal in three branches connected to the junction. The quarter-diurnal signal can be detected, but is not dominant, because the signal energy is weak and discontinuous. This happened because the shallow water effect in the branch does not influence tidal distortion of D2, and thereby cannot produce enough energy to transform from D2 to D4 [19,25]. Moreover, the tidal channel junction is located in the middle of estuary, which is still near to the river mouth, implying that the tide needs to travel more in the landward direction to generate D4.

In contrast, the fortnightly tidal signal does not appear in the CWT analysis in the relationship between the tidal discharge and water level—possibly because of two reasons, that is, (i) the time-series data are too short for capturing fortnightly signal; or (ii) freshwater discharge is very low. For comparison with the work of previous researchers, for example, the work of Leonardi et al. [8], the fortnightly signal does not exist in their wavelet analysis over period of 1.5 months, with the freshwater discharge ranging from 110 to 8235 $m^3s^{-1}$. On the contrary, Sassi et al. [3] could recognize the fortnightly tide using longer time-series data over the period of six months, with the freshwater discharge ranging from 3000 to 5000 $m^3s^{-1}$. However, in our case, the time-series data are less than one month, with the freshwater discharge in the Ota estuary during normal condition ranging from 20 to 40 $m^3s^{-1}$. Thus, in the case of the Ota River, the absence of a fortnightly signal in the interaction between the water level and tidal discharge is the result of the short time-series data [19].

Two main questions arise about the characteristic of phase difference at the junction: (i) the first regarding the temporal variation of the phase difference between the two seaward branches (the eastern and western branches). As a follow-up question, what is the dominant factor that influences the phase difference between the two seaward branches? and (ii) is the phase difference influenced by the spring and neap tide?

To answer the first question, we must first determine the phase difference between the two branches. The phase difference between the tidal discharge and water level at the eastern and western branch for semidiurnal are ~78° and ~88.5°, indicating a mixed wave and mimic standing wave, respectively. It is important to note that the phase difference range of 78° to 88.5° implies that the

tidal discharge leads the water level. The discharge phase difference between the eastern and western branches shows that the discharge phase at the western branch leads by ~15 min compared with the discharge phase at the eastern branch. In addition, the phase difference between the tidal velocity and water level for the eastern and western branches is ~71° and ~80°, respectively (figures not shown). Thus, the phase difference between the velocity and water level is consistent with the phase difference between the discharge and water level, where the phase difference in the western branch is larger than that in the eastern branch. We want to investigate further the reason that the phase difference at the western branch is higher than at the eastern branch. According to Savenije [21], the phase difference is strongly affected by the convergence shape effect, rather than river discharge and bottom friction. Accordingly, we quantify the convergence levels between the eastern and western branches using the exponential function form with three important parameters; namely, the mean water depth, river width, and cross-sectional area, as exemplified by Savenije et al. and Cai et al. [21,27].

Figure 10 shows the geometry of both the eastern and western branches. The geometry analysis presented in Figure 10 (right) obviously indicates that the western branch is more convergent compared with the eastern branch, as shown in Figure 10 (left), resulting in a larger phase at the western branch. From the results of phase difference characteristic and the geometry analysis between two seaward branches, it can be inferred that the amplitude of tidal wave at the western branch near the junction is more magnified compared with at the eastern branch [6,7].

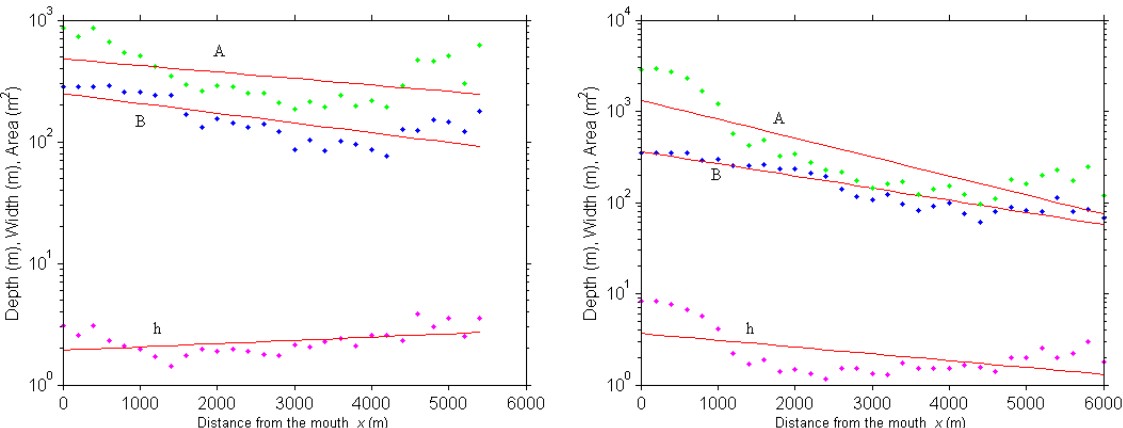

**Figure 10.** Geometry of the eastern branch (left) and western branch (right); showing the longitudinal variation of the cross-sectional area (A), the width (B), and the depth (h).

Figure 11 shows that the phase difference between the tidal discharge and water level at the northern branch (a landward branch of the junction) is ~90°, indicating a standing wave. Thus, it is important to note that the phase difference at the northern branch is slightly higher than that of the two seaward branches. This evidence indicates that the phase difference is slightly increased after passing through the junction into the northern branch. However, further investigation is needed to confirm this evidence.

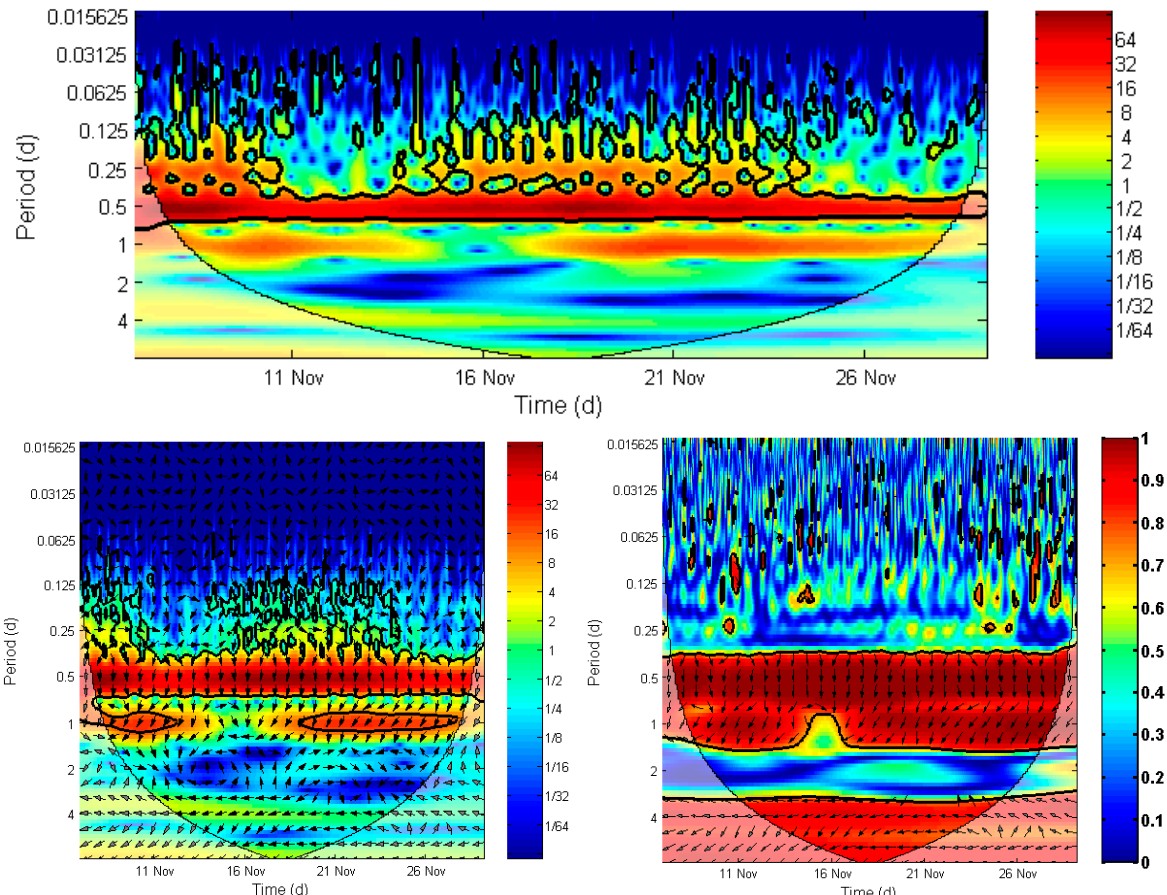

**Figure 11.** Wavelet analyses in the northern branch: (top) continuous wavelet transform; (bottom left) cross-wavelet transform; (bottom right) wavelet coherence. The black contours represent the 0.95 confidence level against red noise, and the cone of influence (COI) where edge effects might distort the picture is shown as a lighter shade. The relative phase relationship is shown as arrows (with in-phase pointing right, anti-phase pointing left, and the first time series leading the second one by 90° pointing down). The wavelet power is $\log_2(A^2/v)$, where $A$ is the wavelet amplitude and $v$ is the variance of the original tidal signals. The $y$-axis in the CWT and XWT is on a $\log_2$ scale.

The answer for the second question is that the effect of spring and neap tide on the phase difference in the semidiurnal domain is negligible. As shown in Figure 9, the phase difference between two seaward branches changes slightly during the spring and neap tide. The phase difference between the tidal discharge and water level is relatively constant, where during the spring tide, the phase difference is ~11°, whereas during the neap tide, the phase difference is ~9°. Therefore, the change of the phase difference between spring and neap tide is not significant. This phenomenon is probably because the phase difference is a function of a ratio between bank convergence and tidal wave length [21]. Moreover, the phase difference can be affected by two factors, that is, the river discharge [8,21] and topography of the estuarine channel, also known as estuary shape [21].

The upstream discharge from Yaguchi gauging station during the experiment is relatively constant, ranging from 20 to 40 m³s⁻¹ during June 2017, though the freshwater discharge from Yaguchi gauging station increased slightly from 20 to 90 m³s⁻¹ in November 2017. Thus, during this condition, the tidal processes of the Ota River can be classified as a tidally dominated estuary [8]. Additionally, this finding is still more or less consistent with the work of Leonardi et al. [8], who pointed out that the phase difference always tends to be close to 90° as long as the characteristic of flow is bidirectional.

## 5. Conclusions

Temporal variations of streamflow were successfully measured by FATS instruments at three branches connected to a tidal channel junction during a low-flow condition. To investigate the flow division of subtidal discharge between the two seaward branches (eastern and western branches), the discharge asymmetry index was used. Moreover, to characterize the tidal discharge behavior and its relationship with the water level and modulated with the freshwater discharge, the phase difference is investigated.

The response of tidal dynamic to the local geometry between two seaward branches shows that the asymmetric discharge at the western branch is more pronounced compared with the eastern branch. The asymmetric tidal discharge at the western branch indicates the ebb discharge is larger than the flood discharge. This happens because the interaction between tidal propagation and the local geometry of western branch that modulated by river discharge could generate the nonlinear effect. This nonlinear effect leads to generation and development of shallow-water constituents, which cause tidal distortion and asymmetry. This asymmetric tidal discharge can induce a seaward sediment movement. In contrast, the characteristic of tidal discharge at the eastern branch is nearly equal during the ebb and flood tide.

Regarding the flow division characteristic at the tidal channel junction of Ota River, the discharge asymmetry index varies from $-0.02$ to $0.26$ during the studied period, suggesting that the eastern branch has the capability to deliver greater amounts of subtidal discharge, approximately 55–63% compared with the western branch, except during the neap tide where the flow division is nearly equal between the eastern and western branch. The temporal change in the flow division is induced by the fluctuation in the subtidal discharge at the eastern branch. The subtidal discharge decreases during the neap tide, so that both subtidal discharges are nearly equal.

The wavelet analyses of the temporal variation of the phase differences between the two seaward branches show slightly different behavior. A mimic standing wave characteristic ($\Delta\varphi = \sim 88.5°$) occurs at the western branch, whereas a mixed wave characteristic ($\Delta\varphi = \sim 78°$) occurs at the eastern branch. It is shown that the phase difference between tidal discharge and the water level is relatively constant during the neap and spring tide. The discharge phase at the western branch leads the discharge phase at the eastern branch by ~15 min. Additionally, the geometry analysis between the two seaward branches reveals that the western branch is more convergent compared with the eastern branch, and thus causes a larger phase difference compared with the eastern branch. As a result, the amplitude of tidal wave at the western branch near the junction is more magnified compared with the eastern branch.

Using the same analysis, we can say the phase difference between the tidal discharge and water level at the northern branch (a landward branch of the junction) indicates a standing wave characteristic ($\Delta\varphi = \sim 90°$), which is slightly higher than the phase difference in the two seaward branches. This evidence implies that the phase difference is slightly increased after passing through the junction into the northern branch. Further study is needed to explore the characteristics of the phase difference in the larger area of the multi-channel estuary.

Although this study deals with the Ota River as a case-specific example, the presented results and discussion are not only aimed at demonstrating the dynamic of flow division and relationship between tidal discharge and water level at a junction, but also take advantage of utilizing the innovative hydro-acoustic system (FATS) for measuring discharge in high temporal resolution and exploring the hydrodynamic process at a tidal channel junction. Moreover, our findings should increase the knowledge for common hydrodynamic processes in multi-channel estuaries that have nearly similar environments over the world.

**Author Contributions:** Conceived the study, M.M.D.; Data curation, M.M.D. and M.B.S.; Validation, M.M.D.; Supervision, K.K.; Writing—original draft, M.M.D.; Writing—review & editing, M.B.S. and K.K.

**Funding:** This work was funded by JSPS KAKENHI, grant number JP17H03313.

**Acknowledgments:** The authors are grateful to Noriaki Gohda of the Hiroshima University/Aqua Environmental Monitoring Limited Liability Partnership (AEM-LLP) for his technical support. Our thanks are also addressed to the members of the Department of Civil and Environmental Engineering, Hiroshima University, for their assistance with the collection of field data in the estuary.

**Conflicts of Interest:** The authors declare no conflict of interest. The funders had no role in the design of the study; in the collection, analyses, or interpretation of data; in the writing of the manuscript; and in the decision to publish the results.

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
