# Peer review of "Characteristics of Tidal Discharge and Phase Difference at a Tidal Channel Junction Investigated Using the Fluvial Acoustic Tomography System"

_water, doi:10.3390/w11040857_

Reviewer 1 Report

I think the paper has significantly improved with respect to the first version. The authors now make clearer what is new and put their results in broader scientific context. The newest aspect for me is the application of the new method to measure discharge. The results on subtidal discharge division confirm earlier results and the discussion of the phase difference between the different channels could be new, but the discussion is relatively limited. Therefore, I have still quite some suggestions to finalize the paper.

It is claimed that the study of phase difference between water levels and discharge, and how it varies over the three channels is new. Actually, it was also studied by Buschman et al. (2013). Furthermore, because the phase of water level is equal at the junction, this study is actually about the phase difference in discharge between the three channels at the junction. Because they have the results of the Wavelet analysis, they can present the phase difference between the Eastern and Western channel as function of time for the D2 tide. Is there a difference between spring and neap? And why would that be? Figure 9 is not very clear in presenting the time variation… Or discuss the values in the text. There seems to be variations in time…

The new text is not always of good quality:

Lines 66 – 71: in my view it is better to remove these lines. For example, I don’t see how phase difference influences damping of tides. And in Line 69: how can the phase difference change something into unidirectional flow?

Line 72: what is meant with ‘variation of phase difference’? not clear whether it is space or time.

Line 72-73: ‘Therefore, to…. Tidal junction.’ Sentence is repeating previous one; can be removed.

Lines 74 – 76: this is a bold statement without any arguments why it is so important. I suggest to remove the sentence.

Line 84:’features of flow’ -> can you be more specific? This is very vague.

Line 84: connected to -> change into ‘connects to’ or ‘are connected to’.

Line 85: Why not change ii) into: Identifying the phase difference in tidal discharge in the channels that connect at the junction.

Line 88: ‘variation in the phase difference’ -> in time or space?

Line 89: I don’t see the importance of issue ii). It is not really discussed in the paper.

 Line 90: The third issue is interesting, but the plot that is used to discuss this is not really revealing..

Line 478: ‘This happens as a result of the interaction between tidal propagation with the local geometry of branch, causing the asymmetry characteristic of tidal discharge in the western branch’ But how does this happen. This does not seem an explanation to me. Which processes are at work that cause this?

Line 486: ‘This can happen because the higher subtidal discharges over spring tide induced by the higher discharge amplitudes at the spring tide with respect to the neap tide’ This needs further explanation. It is not very clearly written.

Author Response

1.      I think the paper has significantly improved with respect to the first version. The authors now make clearer what is new and put their results in broader scientific context. The newest aspect for me is the application of the new method to measure discharge. The results on subtidal discharge division confirm earlier results and the discussion of the phase difference between the different channels could be new, but the discussion is relatively limited. Therefore, I have still quite some suggestions to finalize the paper.

Our reply:

Thank you for your comment. We really appreciate the provided recommendations and suggestions to improve the quality of our paper and hence to be recommended for the publication.

2.      It is claimed that the study of phase difference between water levels and discharge, and how it varies over the three channels is new. Actually, it was also studied by Buschman et al. (2013). Furthermore, because the phase of water level is equal at the junction, this study is actually about the phase difference in discharge between the three channels at the junction. Because they have the results of the Wavelet analysis, they can present the phase difference between the Eastern and Western channel as function of time for the D2 tide. Is there a difference between spring and neap? And why would that be? Figure 9 is not very clear in presenting the time variation… Or discuss the values in the text. There seems to be variations in time…

Our reply:

Thank you for the comment and question for the influence of spring and neap tide on the phase difference. Based on the plot in Figure 9, the arrows actually shows that the temporal variation of phase difference slightly changes during the spring and neap tide. During the spring tide, the phase difference is ~11°, whereas during the neap tide the phase difference is ~. Therefore, the change of the phase difference between spring and neap tide is not significant. Savenije (2012) pointed out that the phase difference can be influenced by the river discharge and topography of estuarine channel or known as estuary shape. He also emphasized that the phase difference is a function of ratio between bank convergence and tidal wave length. Similarly, the work of Leonardi et al. (2015) also indicates that the spring-neap tide does not affect the phase difference of tidal discharge and water level. They stated that the phase difference can only be influenced by river discharge.

Therefore, we have added additional sentences in Line 448-455 as follows:

“As shown in Figure 9, the phase difference between two seaward branches slightly changes during the spring and neap tide. The phase difference between the tidal discharge and water level is relatively constant, where during the spring tide the phase difference is ~11°, whereas during the neap tide the phase difference is ~. Therefore, the change of the phase difference between spring and neap tide is not significant. This phenomenon probably because the phase difference is a function of a ratio between bank convergence and tidal wave length (Savenije, 2012). Moreover, the phase difference can be affected by two factors, i.e., the river discharge (Leonardi, et al., 2015) and topography of the estuarine channel or known as estuary shape (Savenije, 2012).

The new text is not always of good quality:

3.      Lines 66 – 71: in my view it is better to remove these lines. For example, I don’t see how phase difference influences damping of tides. And in Line 69: how can the phase difference change something into unidirectional flow?

Our reply:

Thank you for the suggestion and we agree with it. We removed the lines 66-71 as suggested.

4.      Line 72: what is meant with ‘variation of phase difference’? not clear whether it is space or time.

Our reply:

Thank you for the correction. I have corrected the sentences as suggested, as follows (Line 67:

“The temporal variation of phase difference.”         

5.      Line 72-73: ‘Therefore, to…. Tidal junction.’ Sentence is repeating previous one; can be removed.

Our reply:

Thank you for the correction. We removed it as suggested.

6.      Lines 74 – 76: this is a bold statement without any arguments why it is so important. I suggest to remove the sentence.

Our reply:

Thank you for the suggestion and we agree with it. We removed the lines 74-76 as suggested.

7.      Line 84:’features of flow’ -> can you be more specific? This is very vague.

Our reply:

Thank you for the correction. We have modified it in Line 80 (new submission), as follows: ‘the temporal variation of flow division.’

8.      Line 84: connected to -> change into ‘connects to’ or ‘are connected to’.

Our reply:

Thank you for the correction. We have corrected it as suggested, in Line 76 (new submission) change into: ‘are connected to’.

9.      Line 85: Why not change ii) into: Identifying the phase difference in tidal discharge in the channels that connect at the junction.

Our reply:

Thank you for the suggestion. I change the sentences as suggested as follows:

 ii) Identifying the phase difference in tidal discharge in the channels that are connected to the junction.”

10.  Line 88: ‘variation in the phase difference’ -> in time or space?

Our reply:

Thank you for the comment. We have corrected it as follows:

‘Temporal variation in the phase difference’.

11.  Line 89: I don’t see the importance of issue ii). It is not really discussed in the paper.

Our reply:

Thank you for the correction. We have removed it as suggested.

12.  Line 90: The third issue is interesting, but the plot that is used to discuss this is not really revealing..

Our reply:

Thank you for the comment. As we mention in the second comment above, we have added additional sentences related to the influence of spring-neap tide on the phase difference based on the plot in Figure 9, in Line 448-455 as follows:

“As shown in Figure 9, the phase difference between two seaward branches slightly changes during the spring and neap tide. The phase difference between the tidal discharge and water level is relatively constant, where during the spring tide the phase difference is 11, whereas during the neap tide the phase difference is 9. Therefore, the change of the phase difference between spring and neap tide is not significant. This phenomenon probably because the phase difference is a function of a ratio between bank convergence and tidal wave length (Savenije, 2012). Moreover, the phase difference can be affected by two factors, i.e., the river discharge (Leonardi, et al., 2015; Savenije, 2012) and topography of the estuarine channel or known as estuary shape (Savenije, 2012).

13.  Line 478: ‘This happens as a result of the interaction between tidal propagation with the local geometry of branch, causing the asymmetry characteristic of tidal discharge in the western branch’ But how does this happen. This does not seem an explanation to me. Which processes are at work that cause this?

Our reply:

Thank you for the question. The propagating tidal wave in shallow and narrow branch, and modulated with the river discharge, can generate the nonlinear effect as a result of interaction with the local topography. This nonlinear effect leads to generation and development of shallow-water constituents which cause tidal distortion and asymmetry (Nidzieko, 2010; Lu, et al., 2015). We already mentioned the reasons in ‘the discussion section’, in Lines 385-390 (in the previous submission). In this new submission, we have added additional sentences in the conclusion, in Lines 475-478 as follows:

This happens because the interaction between tidal propagation and the local geometry of western branch that modulated by river discharge could generate the nonlinear effect. This nonlinear effect leads to generation and development of shallow-water constituents which cause tidal distortion and asymmetry.

14.  Line 486: ‘This can happen because the higher subtidal discharges over spring tide induced by the higher discharge amplitudes at the spring tide with respect to the neap tide’ This needs further explanation. It is not very clearly written.

Our reply:

Thank you for the comment. In the discussion section, we change and modified the description for Figure 6 that related to the subtidal discharge and index asymmetry discharge, in Line 388-390, as follows:

“The fluctuation in the subtidal discharge at the eastern branch is greater than that at the western branch; the subtidal discharge at the eastern branch decreases during the neap tide, so that both subtidal discharges are nearly equal during the neap tide.”

As a consequence of modifying the description in discussion section that related to the Figure 6, we also modified the description in the conclusion, in Line 485-487, as follows:

“The temporal change in the flow division is induced by the fluctuation in the subtidal discharge at the eastern branch. The subtidal discharge decreases during the neap tide, so that both subtidal discharges are nearly equal.”

Reviewer 2 Report

An interesting paper which is thoroughly presented covering hydrodynamic aspects of tidal junctions, as well as dominant parameters that influences flow division. Furthermore, the authors shed light on the relationship between tidal discharge and water level using phase difference. 

The problem is interesting and the paper is thorough and intelligently presented. Overall it is an interesting paper in the area of tidal flow with emphasis on tidal junctions, where seaward branches are connected. 

Author Response

An interesting paper which is thoroughly presented covering hydrodynamic aspects of tidal junctions, as well as dominant parameters that influences flow division. Furthermore, the authors shed light on the relationship between tidal discharge and water level using phase difference. 

Our reply:

Thank you for the comment. We really appreciate it.

The problem is interesting and the paper is thorough and intelligently presented. Overall it is an interesting paper in the area of tidal flow with emphasis on tidal junctions, where seaward branches are connected. 

Our reply:

Thank you for the comment. We really appreciate the Reviewer for his/her careful reading and his/her recommendation for publishing.

This manuscript is a resubmission of an earlier submission. The following is a list of the peer review reports and author responses from that submission.

Round  1

Reviewer 1 Report

The paper presents an analysis of discharges at a tidal junction. They use a fairly new method to determine the discharge as a function of time and have calculated the phase difference between water levels and tidal discharges. Furthermore, they have calculated the discharge asymmetry and show the variations in time. I like the new discharge retrieval method, especially when they would have applied three systems simultaneously, so in the West, East and North branch. The method is nice and promising because you get a higher tempral resolution than with classical methods. However, I do not like the analysis of the results. It does not advance our understanding of these systems and I have some doubts about the validity.

The main objective is presented in lines 71 – 74. But it is not specific. What is unknown? They state in line 55-57 that the phase difference between water levels and discharges near junctions has not received a lot of attention, but why should it receive more attention? Why care? It is not explained. It is too simple to state that phase difference controls/determines the tidal wave type.  I agree it is an important indicator, but the details can be subtle. When phase difference is 90 degrees it is not necessarily the case it is a standing waves, it can still be propagating. See work of C.T. Friedrichs and others. When an estuary is strongly convergent and very shallow the phase difference can be (close to) 90 degrees, but the wave is still propagating.  In addition, data on discharge is presented and not data on flow velocity. Because the system is so shallow there can be a significant phase lag between peak discharge and peak flow velocities.

How were the ebb or flood discharge calculated that were used to make Figure5? Were the presented data for the flood and ebb discharge asymmetry based on averages over ebb or flood period (how to take into account the different timing of ebb and flood between the East and West channel), or was a certain time chosen (but which one)? It is therefore very difficult to interpret the results. Why is the difference between ebb and flood important? The difference between spring and neap tide is not very clear. There seems be quite some daily variation as well. Figure 4 could be improved by combining panels b and d into one; even the water levels could be included in the same subpanel by using a double vertical axis. By showing all information together the reader could see whether there is a time difference in ebb or flood, like shown by Buschman et al. (2013).

In addition, I do not understand why the authors do not analyze nor show the subtidal discharges. They state in line 266 it is too small. It is 20 – 40 m^3/s (line 416), which is about 10% of the tidal discharge, which is not really small. I think the authors should use a Godin filter (or any other method) to calculate the subtidal discharge and determine the tidal variations (as the difference between total and subtidal discharge). Figure 4d suggest to me there is about a 20 m^3/s subtidal discharge in the Western channel. The Eastern channel seems to have a smaller value, but a thorough analysis should reveal it. After the subtidal signal is removed from the original signal the Wavelet analysis of tidal signal could be performed as well. Maybe that improved the results, because I am surprised that the quarter diurnal signal does not really show up in the Wavelet analysis (it is not within the confidence limit). The discharge data in Figure 4d clearly is asymmetric.   

Minor points

Figure 2: there seem to be dunes in the river? Can you comment on it.

Figure 3: There are two point clouds, one for peak ebb and one for peak flood. There is no data for small flow velocities. The linear regression is therefore not very trustworthy. Do you also have data around slack tide?

Can you combine section 3.4 with 3.2? At least present 3.3 after section 3.4. The readers should first know whether the method works, before presenting the results.

Line 236 5f ->4f

Reviewer 2 Report

The manuscript presents the measured water level and flow discharge around a channel bifurcation in the Ota River delta in Japan. Based on the measurements the authors discussed the flow distribution between the two seawards channels, and concluded that the distribution is different during ebb than during flood and it shows a spring-neap variation. They also evaluated the phase-lag between the horizontal tide (discharge) and the vertical tide (water level) in the three branches and found that the tide is close to a standing wave around the bifurcation although some differences between the different branches are found. The presented data set is interesting and the results of the analysis may be relevant for the people dealing with this particular delta. However, I cannot support the publication of the manuscript because it is not clear to me what generic knowledge is presented. Why should a reader not dealing with the Ota River delta read this manuscript? What can he/she learn from the presented work?

Although understandable, the manuscript really suffers from the poor English presentation. For people like me who do not have an Indonesian and/or Japanese background it is difficult to follow some parts of the manuscript. It is not merely a matter of problems with grammar. As an example, the statement “and more prominent during the spring tide than during the neap tide” (line 23-24) is not an implication of what is presented in the previous sentences. Therefore it should not be a part of this sentence starting with “This implies that …”. I am not sure but I assume that the illogical reasoning at many places through the manuscript is (partly) due to the language problem.

Another problem concerns the citations which are not always accurate. As an example, the statement “Cai et al. [8] stated that …” concerns the definitions of standing, progressive and mixed waves. Even though Cai et al. (2012) made the statement these definitions are not from them. I do not think that these authors would be pleased by this citation.